# Respiratory Management of the Preterm Infant: Supporting Evidence-Based Practice at the Bedside

**DOI:** 10.3390/children10030535

**Published:** 2023-03-10

**Authors:** Milena Tana, Chiara Tirone, Claudia Aurilia, Alessandra Lio, Angela Paladini, Simona Fattore, Alice Esposito, Davide De Tomaso, Giovanni Vento

**Affiliations:** 1Unità Operativa Complessa di Neonatologia, Fondazione Policlinico Universitario A. Gemelli IRCCS, 00168 Rome, Italy; 2Department of Woman and Child Health and Public Health, Università Cattolica del Sacro Cuore, 00168 Rome, Italy

**Keywords:** preterm infants, respiratory management, respiratory distress syndrome

## Abstract

Extremely preterm infants frequently require some form of respiratory assistance to facilitate the cardiopulmonary transition that occurs in the first hours of life. Current resuscitation guidelines identify as a primary determinant of overall newborn survival the establishment, immediately after birth, of adequate lung inflation and ventilation to ensure an adequate functional residual capacity. Any respiratory support provided, however, is an important contributing factor to the development of bronchopulmonary dysplasia. The risks correlated to invasive ventilatory techniques increase inversely with gestational age. Preterm infants are born at an early stage of lung development and are more susceptible to lung injury deriving from mechanical ventilation. Any approach aiming to reduce the global burden of preterm lung disease must implement lung-protective ventilation strategies that begin from the newborn’s first breaths in the delivery room. Neonatologists today must be able to manage both invasive and noninvasive forms of respiratory assistance to treat a spectrum of lung diseases ranging from acute to chronic conditions. We searched PubMed for articles on preterm infant respiratory assistance. Our narrative review provides an evidence-based overview on the respiratory management of preterm infants, especially in the acute phase of neonatal respiratory distress syndrome, starting from the delivery room and continuing in the neonatal intensive care unit, including a section regarding exogenous surfactant therapy.

## 1. Introduction

Respiratory failure is a frequent and important clinical condition affecting preterm infants inversely to their gestational age (GA) and remains associated with elevated neonatal morbidity and mortality [1,2].

According to currently available data, respiratory distress syndrome (RDS) affects about 80% of neonates born at 28 weeks GA, and this percentage increases to 90% at 24 weeks GA. About 50–60% of these neonates need surfactant administration.

Widespread atelectasis, inability to obtain a functional residual capacity (FRC), hypoxemia, hypercapnia, and excessive work of breathing are clinical features that ex-press respiratory failure in preterm newborns from the first breath in the delivery room [2]. According to this pathophysiology, the current resuscitation guidelines identify the development of appropriate lung expansion and ventilation i.e., of an adequate FRC after birth as the most crucial objective for newborn survival.

It has been demonstrated, however, that any type of ventilation, especially invasive mechanical ventilation (MV), is associated with a major risk of lung injury and consequent bronchopulmonary dysplasia (BPD). Any approach aiming to reduce the global burden of preterm lung disease must therefore focus on the development of lung-protective ventilation strategies.

In order to reduce associated morbidities, the latest resuscitation guidelines emphasize that noninvasive respiratory support should be preferred when treating all respiratory disorders in spontaneously breathing preterm infants.

It is well recognized that the risks correlated to intubation increase inversely with GA. Preterm infants in our era are born at increasingly earlier stages of lung development, and for this reason, they are ever more susceptible to the lung injury deriving from MV.

A current challenge for the neonatologist is to be able to manage both noninvasive and invasive techniques for treating a wide spectrum of lung diseases, ranging from acute to chronic conditions. The respiratory management of a preterm infant must begin from their first breath in delivery room and carry on throughout their entire stay in the neonatal intensive care unit (NICU).

In each paragraph of this narrative review, we have reported the guidelines and the best available evidence, including underlying pathophysiological mechanisms.

The aim of this narrative review is to summarize current evidence on the respiratory management of preterm infants, providing a practical guide for the neonatologist, especially in the early phase of neonatal respiratory distress syndrome (RDS), including these relevant sections:Respiratory management of the preterm infant in the delivery room;Noninvasive respiratory support of the preterm infants in the neonatal intensive care unit;Mechanical ventilation of the preterm infants in the neonatal intensive care unit;Exogenous surfactant therapy in preterm infants.

## 2. Methods

We searched for literature on PubMED to identify the articles that were included in the selection for our narrative review. The research has only been done on Eng-lish-language sources. Using the age "preterm newborn" filter, we restricted the search and utilized the following search terms: “preterm infant AND respiratory management in the delivery room”, “preterm infant AND non-invasive ventilation”, “preterm infant AND respiratory support”, “preterm infants AND nasal continuous positive airway pressure”, “preterm infants AND high flow nasal cannula”, “preterm infant AND heated humidified high flow nasal cannula”, “preterm infant AND Nasal Intermittent Positive Pressure Ventilation”, “preterm infants AND mechanical ventilation”, “preterm infants AND high frequency oscillatory ventilation”, “preterm infants AND Lung-Protective Ventilation”, “preterm infants AND optimal lung volume strategy”, “preterm infants AND Volume-Targeted Ventilation”, “preterm infants AND Weaning from mechanical ventilation”, “preterm infants AND and extubation from mechanical ventilation”, “preterm infants AND exogenous surfactant therapy”.

The final search has been updated to December 2022, and there is no restriction on the year of publication. Also, reference lists of potential publications have been thor-oughly examined to find any material that could have been overlooked throughout the literature search.

Moreover, we included information from the last edition (7th) of Goldsmith’s Assisted Ventilation of the neonate.

## 3. Respiratory Management of the Preterm Infant in the Delivery Room

The majority of preterm infants breathe on their own and only need care throughout the transition from intrauterine life rather than immediate intubation, in contrast to certain preterm infants who are born in poor clinical condition and require resuscitation [1].

The clearance of lung fluid and the establishment of appropriate FRC are the main variables influencing a successful transition in preterm infants.

Preterm infants, however, may struggle to maintain adequate alveolar aeration following the early phase of the transitional process, due to poor inspiratory effort, weak intercostal muscles, poor diaphragmatic function and surfactant deficiency.

In order to facilitate this process, the neonatologist must know the optimal ventilation and oxygen supplementation strategies to adopt in the delivery room.

### 3.1. Ventilation in the Delivery Room

Nasal Continuous Positive Airway Pressure (nCPAP) initiated in the delivery room compared with intubation reduces death and BPD in very preterm infants [3]. Early initiation of CPAP may reduce the duration of MV and postnatal corticosteroid therapy.

The optimal level of CPAP required is unknown, but most studies have used levels of at least 6 cm H_2_O with some as high as 9 cm H_2_O. The reported level of CPAP typically used in delivery rooms is 5 cm H_2_O [4], but infants who do not have a fully established FRC are likely to benefit from higher levels. Current European guidelines advise using a CPAP of at least 6 cm H_2_O [5].

Recently, several studies have emphasized that the use of a single CPAP level from birth until NICU admission may not be optimal, suggesting that a stepwise increase in CPAP according to the patient’s cardiopulmonary response may be more promising [6]; a large study to investigate this is underway [7].

Nevertheless, there is inadequate data from clinical studies to assess the efficacy and safety of sustained lung inflation (SI), which uses greater pressures of up to 20–25 cm H_2_O for a period of around 10–15 s at the beginning of breathing. Therefore, initial SI is not advised for newborn babies who have received Pressure Positive Venti-lation (PPV) [8].

Available evidence suggests starting gentle positive pressure lung inflations with 20–25 cm H_2_O Peak Inspiratory Pressure (PIP) in persistently apneic (no respiratory effort) or bradycardic (heart rate < 100 beats per minute) infants [5]. This should be performed with Positive End Expiratory Pressure (PEEP) to facilitate rapid aeration of the lungs [9].

Endotracheal intubation should be reserved for infants who do not develop adequate respiratory effort and/or who remain bradycardic and/or hypoxic despite adequate PPV via mask or nasal prongs [5,9].

The use of a T-piece significantly decreases intubation rates and the level of maximum pressure applied as well as maximum pressure variability [10]. Therefore, the use of T-piece resuscitators in the delivery room is a better choice than a self-inflating anesthetic bag when compressed gases are available.

### 3.2. Oxygen Supplementation in the Delivery Room

Nearly all preterm infants born ≤32 weeks of GA will require oxygen supplementation in the first 5 min of life to achieve currently recommended oxygen saturation (SpO_2_) targets [9,11].

Current data recommend using a blender to control oxygen supplementation in the delivery room. Fraction of inspired oxygen (FiO_2_) adjustments should be guided by SpO_2_ values measured by pulse oximetry attached on the right hand/wrist.

Monitoring SpO_2_ by pulse oximetry and heart rate by electrocardiography is recommended [9].

Current evidence suggests using an initial FiO_2_ of 30% in preterm infants born <28 weeks of GA [5], and to start with a FiO_2_ between 21 and 30% for those born between 28 and 31 weeks of GA [12]. A small randomized controlled trial (RCT) noted that a higher initial supplemental oxygen concentration was associated with a better respiratory drive and a shorter period of mask ventilation compared to 21% oxygen [13].

A clear recommendation, in addition to promptly starting SpO_2_ monitoring after birth, is to modulate FiO_2_ to achieve SpO_2_ > 80% by 5 min of life, since there is a demonstrated association between hypoxia at this time point and increased mortality [14].

## 4. Noninvasive Respiratory Support of the Preterm Infants in the NICU

Most extremely preterm infants require ventilatory support in the first days of life. Prolonged invasive MV is associated with an increased risk of developing BPD and other comorbidities [15].

Avoiding MV, if possible, and optimizing noninvasive respiratory support in preterm infants reduces lung injury and improves neonatal outcomes [16,17].

Thanks to a deeper understanding of BPD pathophysiology and technological advancements, the use of noninvasive ventilation (NIV) has progressively increased, both as initial support immediately after birth (primary mode) and as post-extubation support for newborns with respiratory failure (secondary mode) [18].

Several noninvasive respiratory support techniques are currently available.

They can be classified into two categories: (1) NIV techniques that provide a constant pressure to the airways during the respiratory cycle, such as CPAP and High Flow Nasal Cannula (HFNC); and (2) NIV techniques that provide a variable pressure to the airways, such as Bilevel Positive Airway Pressure (BiPAP), Nasal Intermittent Positive Pressure Ventilation (NIPPV) and Nasal High Frequency Oscillatory Ventilation (NHFOV) [19].

NIV modes are summarized in Figure 1.

### 4.1. Nasal Continuous Positive Airway Pressure (nCPAP)

Nasal Continuous Positive Airway Pressure (nCPAP) provides continuous positive pressure to the infant’s airways throughout the respiratory cycle, using a noninvasive interface. It allows optimizing FRC and stabilizing the airways, reducing upper airway resistance, obstructive apnea, and alveolar collapse. It also decreases the work of breathing and improves the ventilation–perfusion ratio by decreasing intrapulmonary shunting [20,21,22,23].

CPAP may also stimulate the Hering–Breuer reflex, improving respiratory drive and result in more regular breathing [20,21,22,23].

Major adverse effects of CPAP include: gastric distension and feeding intolerance, nasal injury, and pneumothorax (PTX) [20].

#### Two Main Categories of CPAP Devices Should Be Described

CPAP delivered through a double tube circuit, the so-called “continuous-flow system”. This system uses a flow resistance placed at the end of the circuit to produce a pressure greater than atmospheric pressure. The resistance can be provided from the expiratory valve of a conventional ventilator or by placing the end-expiratory portion of a breathing circuit in water (Bubble-CPAP).CPAP delivered through a single tube circuit, the so-called “variable-flow system”. In this system, two injector jets generate a positive pressure near the interface (Infant Flow Driver). The expiration is facilitated by the Coanda effect [24].

Variable-flow systems seem to be more effective in providing CPAP and reducing the imposed extrinsic work of breathing when compared to continuous-flow systems. Due to the lower resistive work, these systems are particularly preferable in infants diagnosed with BPD [24].

The main interfaces available are short binasal prongs and nasal masks. Nasal masks seem to decrease the risk of CPAP failure (i.e., the need for MV within 72 h from the beginning of CPAP) and reduce the incidence of moderate to severe nasal trauma [25,26,27].

There are not enough data from RCTs regarding the best nCPAP level to use in preterm infants; however, there seem to be potential benefits from using “moderate–high” nCPAP levels, >5 cm H_2_O [28].

As previously described, early use of CPAP in the delivery room compared with elective intubation in preterm infants reduces the risk of respiratory failure, need for MV, and need for exogenous surfactant [21]. A systematic review and meta-analysis including four RCTs (a total of 2782 neonates) showed a decrease in combined outcome of death or BPD in the nCPAP group (RR 0.91, IC 95% 0.84–0.99, NNT 25) [3].

CPAP is indicated as a modality of NIV following surfactant administration via INSURE, INRECSURE or LISA techniques, with a minimal but significant beneficial effect on BPD prevention compared to early MV and a number needed to treat (NNT) of 35 [29].

Post extubation, nCPAP decreases extubation failure compared with oxygen therapy with an NNT of 6 [22].

### 4.2. High-Flow Nasal Cannula (HFNC) Oxygen Therapy

High-Flow Nasal Cannula (HFNC) oxygen therapy is a noninvasive respiratory support system based on the use of a mixture of air and O_2_ delivered at a flow greater than 2 L/min (i.e., greater than the normal inspiratory flow of the newborn). In addition to delivering oxygen, HFNC treatment also provides a distending airway pressure. It is considered an alternative form of NIV to nCPAP that is more manageable both for the newborn and for the caregivers, and it is associated with less nasal trauma. An important consideration when using HFNC treatment is that it is not possible to measure the level of pressure applied to the airways. The gap between the nasal prongs and the nasal cavity during HFNC treatment, as well as other factors such as the size of nasal prongs, gas flow, tracheal diameter, air leakage, and newborn body weight, make it difficult to precisely evaluate the pressure generated by HFNC. Consequently, the airway pressure cannot be well controlled when using HFNC oxygen therapy [30].

In recent years, many studies have been conducted to investigate the efficacy of HFNC compared to nCPAP both as primary respiratory support at birth and as secondary post-extubation support [31,32,33,34,35,36]. HFNC is frequently used as respiratory support in the NIV weaning phase as a stepdown mode from nCPAP [37,38,39].

In recent years, many reviews and meta-analyses have been published with mixed results. RCTs evaluating the use of HFNC as respiratory support in preterm infants and its efficacy compared to nCPAP were included [40,41,42,43,44,45].

Recently, a systematic review and meta-analysis has been published with the aim to compare HFNC versus nCPAP as primary respiratory support in preterm infants to provide evidence-based support for clinical practice [46]. The primary outcome considered was the incidence of treatment failure and the need for MV after non-invasive respiratory support. RCTs included in the meta-analysis resulted from a search conducted in the main databases until February 2022, without language restriction. At the end of the research, twenty-seven studies were included for a total of 3351 newborns (1664 in the HFNC group; 1687 in the nCPAP group). Concerning primary outcomes, the meta-analysis of the results from twenty-two studies reporting the incidence of respiratory support failure did not show a significant difference in this outcome.

The same result emerged for the application of MV after non-invasive respiratory support by evaluating twenty-one studies reporting this outcome. When four studies [6,7,47,48] were analyzed for the application of alternative noninvasive respiratory support when treatment failure occurred, no difference resulted in the application of MV between the HFNC group and CPAP group. This concludes that using nCPAP as a remedy for the treatment failure of HFNC does not avoid intubation, and the intubation rate difference is insignificant once failed cases on HFNC are rescued by nCPAP.

Moreover, concerning secondary outcomes, the HFNC group showed a later onset of MV, lower duration of oxygen therapy, earlier initiation of enteral feeding and lower rate of air leaks, nasal trauma, and abdominal distension.

It is important to consider that the RCTs included in this recent meta-analysis predominantly enrolled preterm infants with GA > 28 weeks. Only three studies [49,50,51] clearly indicate enrollment of infants with GA < 28 weeks (extremely low gestational age newborns, or ELGANs). Among these, one study [51] indicates the enrollment of newborns between 26 and 34 weeks of GA, without however providing precise information about the characteristics of the participants. The other two studies did not show significant differences in safety and efficacy between HFNC and nCPAP [49] with a lower frequency of intubation in neonates of the HFNC group [50]. Because of the lack of specific data on individuals, subgroup analysis based on GA or birth weight (BW) could not be performed. Thus, for ELGANs, the use of HFNC as primary respiratory support still needs to be further clarified.

When the use of HFNC is compared to nCPAP as secondary respiratory support after extubation following a period of MV in preterm infants, HFNC is associated with more treatment failure than nCPAP, but nCPAP has a significantly increased rate of nasal trauma and PTX [43].

Concerning the use of HFNC as a method to improve the weaning from nCPAP, a study by Badiee et al. [37] published in 2015 showed that weaning from nCPAP to HFNC could decrease the duration of oxygen therapy and length of hospitalization in preterm infants. In this study, infants in the HFNC group received HFNC with a flow of 2 L/min and a FiO_2_ of 30% and then were subjected to progressive stepwise reduction in FiO_2_ and then flow. On the other hand, the non-HFNC group was maintained on an nCPAP of 5 cm H_2_O and gradually reduced oxygen until reaching a FiO_2_ equal to 21% for 6 h; then, they were directly weaned from nCPAP (with pressure of 5 cm H_2_O) to room air. The study was conducted several years ago when the use of HFNC with a flow > 2 L/min was not yet widespread.

Further studies are needed to define the best method of weaning from nCPAP, since to date, there has been no agreement on the optimal approach.

Another important aspect is NIV weaning in infants diagnosed with BPD.

Wright et al. address this topic by providing operational indications [52]. These authors propose to start NIV weaning only when the infant can maintain stability with no respiratory support or with only low oxygen flow when awake and requires NIV only during sleep (both night and naps) for a maximum of 16 h/24 h a day. They progressively suspend respiratory support when the newborn is awake by increasing the duration of the suspension phases and ensuring stability during sleep until subsequent definitive night-time suspension of respiratory assistance can be achieved.

This method allows the newborn to be free from respiratory aids during waking phases, with a consequent better interaction with the surrounding environment. These authors suggest a flow reduction of 0.5–1 L/min every 24 h down to a minimum of 2–3 L/min.

HFNC has been recently proposed as support in infants with severe BPD only if previously supported in CPAP with a pressure value ≤ 6 cm H_2_O and FiO_2_ ≤ 0.25. Generally, infants with established BPD require a starting flow rate > 6 L/min and sometimes up to 8 L/min [38].

In this category of newborns, a period of stability of at least 24–48 h is considered necessary before proceeding with the reduction in flow by 1 L/min. The suspension should be considered when a flow of 2 L/min is reached [38].

Overall, no studies have been conducted on the best way of weaning CPAP or HFNC in infants with established severe BPD, and only generic indications based on the experience of each neonatal care center have been provided.

### 4.3. Nasal Intermittent Positive Pressure Ventilation (NIPPV)

Nasal Intermittent Positive Pressure Ventilation (NIPPV) assists the newborn’s breathing with the same principles of Conventional Mechanical Ventilation (CMV): two pressure levels keep the airways open (PIP and PEEP) and the inspiratory time and the ventilator rate define the frequency and duration of each phase [53]. Intermittent positive pressure can be generated by a ventilator (CMV-NIPPV) or by a flow driver (BiPAP). In the second case, PIP is lower and longer [54].

NIPPV provides the same benefits as CPAP. In addition, it supports tidal ventilation and increases the main airway pressure, improving alveolar recruitment.

The interfaces are the same as those described for CPAP. Adverse effects are also similar among the two techniques, with the same risk of gastrointestinal problems. A reduced risk of PTX was, however, observed with NIPPV compared to CPAP [55].

NIPPV can be used as a primary mode of ventilation and following surfactant administration with the INSURE/INRECSURE techniques. In this case, compared to CPAP, it reduces the need for MV (RR 0.78, 95% CI 0.64–0.94) and reduces the need for a second dose of exogenous surfactant administration [56]. These findings of the 2016 Cochrane Review are consistent with those of a more recent meta-analysis conducted by Rüegger CM et al. [57].

The aim of these authors was to evaluate respiratory failure leading to additional ventilatory support in preterm infants with respiratory distress syndrome (RDS), less than 6 h old, and supported with NIPPV versus nCPAP. By pooling data from eighteen trials, the meta-analysis demonstrated a clinically important 37% relative reduction in the risk of respiratory failure with NIPPV compared to nCPAP.

Comparing subgroups of infants, the beneficial effect resulted to be more obvious in the trials using ventilator generated NIPPV and was even greatest when synchronization was used (sNIPPV). With ventilator generated sNIPPV, the NNT to prevent one respiratory failure was 7.

When used as a secondary mode following a prolonged period of MV, NIPPV has been found to be more effective than CPAP in preventing extubation failure in infants recovering from RDS (RR 0.70, 95% CI 0.60–0.80, NNT 8; RR 0.76, 95% CI 0.65–0.88, NNT 10), with a major benefit when NIPPV is generated by a ventilator (RR 0.51, 95% CI 0.40–0.65) [57,58].

NIPPV can be synchronized with the infant’s spontaneous breathing thanks to devices capable of identifying the patient’s respiratory effort by detecting variations in pressure or flow. The most common methods for NIPPV synchronization are the Graseby capsule and airflow sensors.

Using the airflow signal as a trigger signal has the advantage of ensuring that the mechanical cycle is delivered when the vocal cords are open.

Theoretically, the asynchrony between the spontaneous breaths and the breaths delivered by the ventilator can induce laryngeal closure, inhibit inspiration, increase abdominal distension, and have negative effects on systemic pressure and cerebral blood flow as well as increase the work of breathing. Such effects may be overcome with synchronization.

Studies on preterm infants indicate that when compared with CPAP, sNIPPV reduces the work of breathing, reduces thoracoabdominal asynchrony, increases tidal volume and minute ventilation, and reduces CO_2_ concentration [59,60,61].

There are few studies comparing sNIPPV and NIPPV, and the results are conflicting. According to these studies, sNIPPV seems to reduce the incidence of apnea in preterm infants [62], reduce respiratory effort, improve infant–ventilator interaction [63] and overall increase the benefits of NIPPV [55]. However, a retrospective study found no significant impact on clinical outcomes in the use of sNIPPV compared with NIPPV [64]. A recent meta-analysis shows that sNIPPV is the most effective mode when used as post-extubation support [65].

### 4.4. Practical Tools of Noninvasive Respiratory Support in the NICU

To date, based on the available evidence, the most effective noninvasive respiratory support as the primary mode is NIPPV, especially when delivered by a ventilator and with synchronization (i.e., sNIPPV).

As an alternative primary support, it seems appropriate to choose nCPAP in ELGANs (infants born < 28 weeks of GA).

Preterm infants with more than 28 weeks of GA do not show significant differences when nCPAP is compared to HFNC. In this category of preterm infants, the later onset of MV, lower duration of oxygen therapy, earlier initiation of enteral feeding and lower rate of air leaks, nasal trauma and abdominal distension when treated with HFNC should therefore be taken into consideration

As secondary respiratory support after extubation following a period of MV in preterm infants, the use of HFNC instead of nCPAP should not be taken into consideration as HFNC is associated with more treatment failure than nCPAP.

Again, according to current evidence, it seems reasonable to choose NIPPV as post extubation respiratory support in preterm neonates at high risk of extubation failure, especially in ELGANs.

Subsequently, when the weaning criteria are satisfied, it will be possible to progressively reduce respiratory assistance and shift it to nCPAP. Based on these data, an algorithm on the choice of noninvasive respiratory support as secondary mode, based on GA and the presence of certain risk factors of extubation failure, can be proposed as shown in Figure 2 (modified from Shehaded AMH et al.) [66,67,68,69].

## 5. Mechanical Ventilation of Preterm Infants in the NICU

As discussed above, the current recommendations suggest the primary use of noninvasive respiratory support in spontaneously breathing preterm infants to avoid MV, which has been associated with increased mortality and pulmonary and systemic morbidities [5,70,71].

The number of preterm infants treated with noninvasive respiratory support is in fact increasing, but a substantial proportion of these (almost 50–70%), especially ELGANs, (i.e., <28 weeks of GA), continue to require lifesaving intubation and MV in the delivery room and/or at some point during their NICU stay [71,72].

The risk of intubation and MV increases inversely with GA: the preterm infants who receive MV today tend to be very small and uniquely susceptible to lung injury because of the very early stage of lung development present at birth.

As a result, they may become ventilator-dependent for extended periods of time, even for conditions unrelated to their lung disease [71,72].

For these reasons, the neonatologist today must acquire the skills necessary to manage MV for a spectrum of lung diseases that range from acute to chronic conditions.

In preterm infants requiring MV, the respiratory support must be optimized according to the best evidence-based and consensus practices available for the specific lung disease treated to reduce the rate of mortality and morbidity related to MV [71,73].

### 5.1. Indications for Mechanical Ventilation in Preterm Infants

To the best of our knowledge, there are still no evidence-based standardized indications regarding the intubation and MV of preterm infants, also in consideration of their wide range of clinical conditions, gestational ages and birth weights. Reasonable practice indications are summarized in Table 1.

### 5.2. Goals of Mechanical Ventilation in Preterm Infants

The main objectives of MV are to optimize gas exchange, minimize adverse effects (i.e., acute lung injury, ventilator-induced lung injury (VILI), air leak syndrome, airway damage, hemodynamic impairment, nosocomial infection, and brain injury), ensure comfort by reducing asynchrony, work of breathing and oxygen consumption, and early weaning from invasive support by attempting extubation as soon as possible [71,73].

To reduce incidence of BPD and other morbidities, available evidence continues to support the need for weaning from MV during the first week of life, considering a trial of extubation in infants who tolerate weaning to low settings (Mean Airway Pressure (MAP) ≤ 8 cm H_2_O and FiO_2_ ≤ 0.30), even if long-term success is not guaranteed [73,74,75].

Suggested goals of MV in preterm infants during NICU stay are reported in Table 2.

#### 5.2.1. Oxygenation and Use of Supplemental Oxygen

It is known that preterm infants are very susceptible to both hyperoxic and hypoxic states. Hyperoxia can contribute to compromise the developing lungs and retina, while hypoxia can lead to increased mortality, cerebral white matter lesions and necrotizing enterocolitis (NEC).

Several studies have indicated that preterm neonates have impaired ability to increment antioxidant enzymes in conditions of hyperoxia, and this determines important vulnerability to oxidative stress [76,77].

For this reason, a series of RCTs and meta-analyses have tried to define the optimal oxygen saturation range for preterm infants requiring oxygen supplementation in the NICU after postnatal stabilization, but results remain unclear [78,79].

While waiting for new evidence, a reasonable approach seems to be to maintain preterm infants requiring supplemental oxygen during their NICU stay within an SpO_2_ range of 90% to 95% [73,77].

This goal needs to be obtained using the lowest FiO_2_ possible and optimizing oxygenation using adequate ventilation/perfusion matching as well as optimizing End-Expiratory Lung Volume (EELV) with MV. PEEP adjustments are the most effective ways of optimizing EELV during CMV. Direct adjustments in MAP optimize lung volume and ventilation/perfusion matching during HFOV.

Another important concept is to avoid both prolonged periods of hypoxia (SpO_2_ < 80%, associated with neurodevelopmental delay) and fluctuations in SpO_2_, because the alternance of hyperoxia and hypoxia represents a proinflammatory stimulus.

Recently, the automatic control of FiO_2_ to improve oxygen saturation targeting has become a promising tool to avoid these damages [77].

#### 5.2.2. Ventilation (CO_2_ Elimination)

In ventilated preterm infants, ventilation (i.e., CO_2_ elimination) needs to be optimized in order to achieve an acceptable range of pH and pCO_2,_ avoiding hypocapnia (pCO_2_ < 35 mmHg) to decrease the risk of cerebral vasoconstriction and accepting a mild permissive hypercapnia (pCO_2_ ≤ 55 mmHg).

However, it is important to avoid PaCO_2_ > 60 mmHg in the first 72 h of life due to the greater risk of cerebral vasodilatation and consequent intraventricular hemorrhage (IVH) [71,80].

Some clinicians accept this higher PaCO_2_ value in ventilated preterm infants to reduce tidal volume as much as possible, but while experimental evidence suggests its protective effect on the lungs, studies in ventilated preterm infants did not show a clear benefit in terms of BPD reduction [81,82].

An important concept is to avoid oscillations and rapid changes in PaCO_2_ values as much as possible.

### 5.3. Practical Suggestions and Monitoring during Mechanical Ventilation in Preterm Infants

Although MV may be lifesaving in preterm infants, it is associated with many possible adverse effects and requires a thorough understanding of respiratory physiology and a familiarity with the ventilators [83].

Once MV has begun, it is necessary to choose the most appropriate ventilation modality and strategy based on the pathophysiology of the treated lung disease and on the specific NICU skills. It has been demonstrated that unit protocols and guidelines are useful, but optimal outcomes can be achieved only individualizing patient care with frequent assessments of the patient’s response to ventilator settings [84].

During MV, all preterm infants must be monitored by continuous cardiorespiratory monitoring, continuous pulse oximetry (SpO_2_), and at least intermittent blood pressure and temperature monitoring.

An arterial catheter for continuous monitoring of blood pressure and periodic arterial blood gas sampling is highly desirable in the unstable and critically ill preterm infants, especially ELGANs.

It is recommended to carry out appropriate monitoring of blood gases (a sample of blood gases should be analyzed within 30–60 min from the start of ventilation) and/or monitoring of transcutaneous pO_2_ and pCO_2_, to verify the appropriateness of the selected parameters and make any small changes that allow for the greatest possible stability of pO_2_ and pCO_2_ values, especially in conditions of rapid change in lung mechanics (e.g., surfactant administration).

Cerebral tissue oxygen saturation by near infrared spectroscopy (NIRS) monitoring is desirable.

Ventilation settings and measured variables should be recorded at regular intervals.

The humidifier temperature should be regularly checked and recorded.

Evaluating the infant’s clinical response and real-time lung graphs provided by most modern ventilators can be very helpful in fine-tuning ventilator settings.

The need for radiodiagnostic examinations should be assessed (e.g., lung ultrasound, chest X-ray).

### 5.4. Physiologic Concepts of Respiratory Failure, Ventilator-Induced Lung Injury and Lung-Protective Ventilation in Preterm Infants

Whilst MV in preterm infants with respiratory failure may be lifesaving, it is well known that it may result in a form of severe lung damage termed Ventilator-Induced Lung Injury (VILI). VILI represents one of the major risk factors for the subsequent development of BPD [73,85].

The lungs of preterm infants are more susceptible to VILI if antenatal inflammation (e.g., chorioamnionitis), postnatal inflammation (e.g., sepsis, pneumonia), and surfactant deficiency are present at the start of MV [86,87].

Furthermore, several studies have contributed to identifying the risk factors for VILI and its mechanisms in preterm infants, helping to develop so-called “lung protective” ventilation strategies that minimize the risk of VILI associated morbidity and mortality.

A deeper understanding of the pathophysiology of preterm infant respiratory failure and VILI can help to better understand the concepts behind lung-protective ventilation strategies used in preterm infants [80].

#### 5.4.1. Physiological Concepts of Respiratory Failure in Preterm Infants

The lungs of preterm infants are structurally immature and deficient in surfactant, resulting in neonatal RDS in the first days of life.

Lungs of preterm infants, especially ELGANs, are characterized by increased pulmonary elastic recoil forces due to the higher surface tension at the alveolar/saccular air–liquid interface and a concomitant reduction in lung compliance.

The elevated chest wall compliance of the preterm infant is unable to contrast these increased recoil forces, resulting in a reduction and instability of the EELV.

This low EELV may lead to a further reduction in lung compliance and increased airway resistance and work of breathing.

Collapse of saccules may increase intrapulmonary right-to-left shunting, leading to hypoxia and an irregular distribution of tidal volume.

#### 5.4.2. Ventilator-Induced Lung Injury (VILI) and Its Consequence

The most important risk factors for VILI in preterm infants are: volutrauma, atelectrauma and oxygen toxicity.

Volutrauma is associated to alveolar/saccular overdistension, often caused by high tidal volume ventilation. Sometimes, it may be induced by low tidal volumes provided at the airway opening, which result in regional overdistention in atelectatic lungs or by low tidal volumes superimposed on a high EELV that can still exceed total lung capacity [88];Atelectrauma is associated with repetitive collapse and reopening of alveoli/sacculi as a result of surfactant deficiency or inhibition [89];Oxygen toxicity in preterm infants has been shown in several studies demonstrating the inability of preterm infants to increase antioxidant enzymes in response to hyperoxia, resulting in a major vulnerability to oxidative stress [76].

It is important to note that in preterm infants, poor ventilation/perfusion matching, intrapulmonary right-to-left shunting, as well as low EELV all result in high oxygen requirements, thus increasing oxidative stress.

The pulmonary and systemic consequences of VILI are: biotrauma (increased pulmonary and sometimes systemic inflammatory response), structural injury to the alveolar-capillary unit (increased endothelial and epithelial permeability, with loss of fluids and plasma proteins in the alveolar space, causing pulmonary edema and pulmonary hemorrhage), surfactant dysfunction (inactivation and reduced synthesis), and lung development alteration (arrest of the normal alveolarization process) [90,91,92,93].

#### 5.4.3. Lung-Protective Ventilation and Optimal Lung Volume Strategy

Although MV in preterm infants must be tailored to address each patient’s specific conditions and timing of lung disease, current recommendations for MV in preterm infants emphasize the importance of the ventilation strategy over the ventilation mode.

At all times, preterm infants needing MV must be ventilated using a lung protective strategy whose goal is to obtain adequate gas exchange while minimizing VILI.

Based on the cascade of events that determine VILI, the principles of a lung-protective strategy for preterm infants are:Minimize atelectrauma by optimizing EELV. This is achieved by reversing atelectasis using recruitment maneuvers and stabilizing lung units during the ventilatory cycle (by applying sufficient airway pressure at the end of expiration);Minimize volutrauma by reducing alveolar overdistension. This is achieved with limited tidal volumes that must be distributed in a homogeneously aerated lung [80,94,95,96].

Generally, these strategies improve oxygenation and reduce FiO_2_ requirement and consequently oxygen toxicity.

A lung-protective ventilation strategy based on these principles is called “open lung ventilation strategy” or optimal lung volume strategy, and it may be applied both in CMV and HFOV [94].

#### 5.4.4. Practical Tools for Lung-Protective Ventilation in CMV

In order to minimize volutrauma during CMV in preterm infants, experts recommend targeting the tidal volume between 4 and 7 mL/kg.

To date, evidence supporting this range is limited. Further RCTs should evaluate the optimal tidal volume range during CMV for specific lung diseases of preterm infants (i.e., RDS) [70].

Conversely, there is evidence of the use of volume-target ventilation (VTV) stabilizing tidal volume and resulting in a reduction in the risk of BPD, hypocapnia, and other morbidities [97].

Although studies on the open lung ventilation strategy during CMV in preterm infants are limited, the data currently available suggest to consider it in preterm infants affected by RDS, aiming at oxygenation improvement and FiO_2_ reduction [98]. The open lung ventilation strategy during CMV may optimize EELV by adequate levels of PEEP to recruit and stabilize lung units, using oxygenation as an indirect indicator of lung volume (in the absence of methods that allow a direct measurement of lung volume).

To our knowledge, there is not a single optimal level of PEEP suggested. It must be sought on each individual patient through lung recruitment procedures, considering lung compliance and the phase of the lung disease being treated (acute/chronic/weaning) to work on the favorable portion of the pressure/volume curve.

In preterm infants, experts suggest using a PEEP value of at least 5–6 cm H_2_O because physiological lungs are rarely ventilated in these patients.

Further studies will be necessary to confirm the efficacy and positive effects of the open lung ventilation strategy (recruitment and stabilization) during CMV in preterm infants.

#### 5.4.5. Practical Tools for Lung-Protective Ventilation in HFOV

The risk of volutrauma using HFOV in preterm infants is inherently reduced because this ventilation mode, by design, uses very low tidal volumes smaller than the anatomical dead space (i.e., 1.5–2.5 mL/kg).

Nevertheless, animal studies have demonstrated that HFOV may be considered lung protective only if combined with an open lung ventilation strategy [99], because in this way, very small tidal volumes can be applied in a recruited and stabilized lung, with the lowest possible airway pressure.

To minimize atelectrauma during HFOV in preterm infants with RDS, it is recommended to use an oxygenation-guided lung recruitment maneuver by continuous distending pressure (CDP) stepwise increase–decrease and then stabilizing lung units applying an adequate CDP level to place ventilation on the deflation limb of the P/V curve. This strategy provides optimal volumes and minimal distending pressure, avoiding both overdistension and atelectasis [95,96].

Although there is poor evidence regarding the use of HFOV combined with volume guarantee (HFOV-VG) in preterm infants, this strategy seems to improve tidal volume and ventilation stabilization over time, showing better short-term neonatal outcomes. Therefore, HFOV-VG should be suggested during the ventilation of preterm infants, especially in situations with rapid changes in lung mechanics, such as surfactant administration [100,101].

### 5.5. Volume-Targeted Ventilation (VTV) in Preterm Infants

Traditionally, pressure-controlled (PC) ventilation has been the dominant form of ventilation used in the neonatal population as opposed to adult medicine, which preferentially uses volume-controlled ventilation. The main historical reasons for the widespread use of PC ventilation in neonates are the need to ventilate despite large leaks around the uncuffed endotracheal tubes (ETTs) and the fear of using excessive pressures.

Nevertheless, it has been demonstrated that volume, rather than pressure, is the main cause of VILI. Several animal studies have shown that high pressure without high volume is not as injurious as high volume itself [88,102,103].

Indeed, the introduction of VTV has been a big step forward in ventilated preterm infants: VTV allows the clinician to use a PC ventilation and, at the same time, to keep the tidal volume (VT) constant.

VTV modalities are modifications of PC ventilation designed to deliver a target VT by real-time microprocessor-directed adjustments of inflation pressure.

Some devices regulate tidal volume delivery based on flow measurement during inflation and others during exhalation. The most studied modality of VTV is volume guarantee (VG) ventilation, whose algorithm works on exhaled volume.

In VG modality, the operator chooses a target VT and a pressure limit up to which the ventilator operating pressure (working pressure) may be adjusted. The most used algorithm compares the exhaled VT of the previous inflation and adjusts the working pressure up or down to target the set VT. If the ventilator is unable to reach the target VT with the set inflation pressure limit, a “low tidal volume” alarm will sound, alerting the operator that an assessment is needed.

VTV leads to a stable VT thus avoiding large fluctuations of PaCO_2_. Since both hypocapnia and hypercapnia are related to brain injury in the first days of life [104,105], VG-modality could be seen as a “neuro-protective” and not just a “lung-protective” ventilation strategy.

It is common to inadvertently deliver excessive VT using PC ventilation during the recovery phase of RDS. In this tricky temporal window, when lung compliance quickly improves and the brain reaches peak vulnerability, VG modality allows an automatic pressure weaning that keeps ventilation gentle and stabilizes PaCO_2_ values [97].

Two recent meta-analyses that included a combination of several different modalities of VTV documented several advantages of VTV compared with PC ventilation in preterm infants (6,7); these included a significant decrease in the combined outcome of death or BPD, lower rate of PTX, less hypocapnia, decreased risk of severe intraventricular hemorrhage/periventricular leukomalacia, and significantly shorter duration of MV [97,106].

The choice of appropriate VT depends on infant size and pulmonary conditions [107]. The smallest infants require a slightly larger VT/kg owing to the proportionally larger fixed instrumental dead space [108].

In case of pulmonary conditions that lead to increased alveolar dead space (e.g., meconium aspiration syndrome or BPD), a relatively larger VT should be used. As the underlying pulmonary disease evolves and the baby grows, the optimal VT target will also change.

Nevertheless, the weaning process does not require decreasing VT as the physiologic VT required by the patient does not decrease with improving clinical condition: what decreases is the pressure required to achieve that VT because of improved compliance of the respiratory system and the infant breathing more effectively [109].

Other than VT, the VG-modality requires to set a Pmax level, which is the PIP which the ventilator cannot exceed. Pmax should be set 5 to 10 cm H_2_O above the “working PIP” used to reach the set VT. The Pmax should be set to monitor lung condition, as the working PIP rises in case of worsening lung conditions or complications of ventilation such as tube obstruction [109].

Even when using VTV, the open lung ventilation strategy (recruitment and stabilization) should be performed. As previously described, the benefits of VTV occur only if the VT is distributed into an “open lung” since, when gas enters partially atelectatic lungs, the VT will preferentially go to the already aerated portion of the lungs [110]. Based on Laplace’s law, the pressure required to expand the aerated lung is less than the critical opening pressure of the atelectatic alveoli.

In recent years, VG modality has been combined with HFOV (HFOV-VG). HFOV-VG has been demonstrated as a feasible and safe modality in preterm infants with RDS and reduces fluctuations of VT during HFOV (VThf) and pCO_2_ levels compared to HFOV alone [111,112].

Moreover, in preterm infants with RDS, HFOV-VG has been shown to be safer than HFOV alone after surfactant administration by providing greater VThf and pCO_2_ stability despite the quickly improving lung compliance. In this study, an HFOV-VG starting ventilator setting with VThf 1.5–1.8 mL/kg and a frequency of 15 Hz has been demonstrated to be safe and efficacious in extremely preterm infants affected by RDS in the first hours of life [101].

In conclusion, although more studies are needed to confirm evidence and the presumed short- and long-term beneficial effects, the use of VTV or HFOV-VG for ventilated preterm infants should be considered in association to the open lung ventilation strategy.

The selection of VT or VThf may be individualized for the specific patient, considering their lung mechanics, lung dimension and the phase of the lung disease being treated (acute/chronic/weaning) (i.e., low tidal volume during the early phase of RDS vs. high tidal volume during severe BPD).

### 5.6. Weaning and Extubation from Mechanical Ventilation in Preterm Infants

Weaning from MV is the process of gradually decreasing ventilatory support with a simultaneously gradual increase in patient work of ventilation.

Weaning preterm infants from MV as soon as possible represents an imperative for all neonatologists. Despite advances in respiratory care, it is still a challenge to understand whether a preterm infant is ready for extubation or not.

During VTV, the decision is usually taken when gas exchange is achieved with low working-PIP and low FiO_2_ and when the infant shows a regular breathing pattern.

In HFOV-ventilated preterm infants, the extubation may be performed directly without needing a period of CMV. It has already been demonstrated that in Extremely Low Birth Weight (ELBW) infants electively ventilated with HFOV and an open lung strategy, direct extubation from HFOV at MAP ≤ 6 cm H_2_O with FiO_2_ ≤ 0.25 is feasible with a high extubation success rate (83%) [113].

Apparent clinical readiness and minimal ventilator settings before extubation are the main parameters used to decide when to extubate; nevertheless, they are not always reliable.

In the APEX cohort study, neonates were extubated from different settings and parameters, including MAP of 5–14 cm H_2_O, FiO_2_ requirements ranging from 0.21 to 0.53 and pCO_2_ levels from 22 to 69 mmHg. No significant difference was observed in the success of extubation [114].

In adults, spontaneous breathing trials (SBTs) have been proved effective for assessing extubation readiness. In newborns, several studies have tested the efficacy of SBTs, which usually consist in a 3- to 10-min period of spontaneous breathing via endotracheal CPAP (ET-CPAP), during which pass or fail is determined from a combination of clinical events (apnea, bradycardia, and desaturations). According to these studies, SBTs have high sensitivity but low specificity in evaluating the readiness for extubation in newborns [115].

Moreover, in a recent study, 67% of preterm infants exhibited at least one clinical event while undergoing a 5-min ET-CPAP immediately before extubation [116], which is probably due to the increased resistance and dead space of the endotracheal tubes; thus, there are many concerns about the high level of work of breathing during SBTs causing clinical instability in preterm infants even if just for a few minutes [117]. SBTs in premature infants can accurately predict extubation success but not extubation failure. Therefore, even though it is an easy-to-perform bedside assessment tool, there is a lack of evidence to support its use in neonatal medicine [115,116,117].

Currently, research focus has moved onto new technologies which can increase the specificity of SBTs.

Some studies have demonstrated the utility of studying the autonomic nervous system during weaning: Kacsmarele et al. studied Heart Rate Variability (HRV) during the weaning process and found a markedly lower HRV in newborns that eventually failed extubation [118]. This test seemed to have high specificity and sensitivity, but more data are needed to recommend this tool for assessing extubation readiness. Other studies investigated the predictive ability of the Respiratory Variability Index (RVI), finding a lower RVI in those newborns failing extubation [119].

A recent study by Williams EE et al. used transcutaneous electromyography of the diaphragm (dEMG) to study the electrical activity of the diaphragm (Edi) during an SBT to detect extubation readiness in preterm newborns [22]. They found that in babies born at a GA lower than 29 weeks, an increase in the Edi was the best predictor of extubation failure with moderate sensitivity and specificity [120]. Moreover, a study by Vento G et al. described that the spontaneous minute ventilation during an Et-CPAP period before extubation was significantly lower in infants who failed extubation compared to the ones who were successfully extubated [121]. Future works studying both the Edi and minute ventilation during SBTs may improve our understanding of the neuroventilatory efficiency in preterm babies and increase the predicting power of SBTs.

In conclusion, to our knowledge, none of the proposed indicators of extubation readiness were data evidenced.

As we previously discussed, data currently suggest attempting extubation in ventilated preterm infants as soon as possible, preferably in the first week of life, to reduce incidence of BPD and other morbidities.

We suggest considering extubation when the infant is clinically stable: blood gas and SpO_2_ are in the target range with a FiO_2_ less than 30% and an MAP less than 8 cm H_2_O, both on CMV and HFOV.

## 6. Exogenous Surfactant Therapy in Preterm Infants

Surfactant is a naturally produced surface-active lipoprotein complex mixed with proteins, which reduces the surface tension at the alveolar liquid surface, thus allowing alveoli to remain open during expiration and substantially reducing the work of breathing. Surfactant also improves muco-ciliary transport, prevents the formation of pulmonary oedema, improves pulmonary compliance, and contributes to lung defense against pathogens [71,122].

Considering these reasons, exogenous surfactant therapy is one of the most important treatments for preterm infants.

The administration of exogenous surfactant is lifesaving in the management of RDS in preterm infants, especially ELGANs, and it is proving increasingly useful for other neonatal respiratory disorders that feature an altered surfactant homeostasis (i.e., meconium aspiration syndrome, pneumonia, pulmonary hemorrhage, neonatal acute respiratory distress syndrome) [71].

### 6.1. Surfactant Therapy in Preterm Infants with RDS

In the management of RDS in preterm infants, the administration of exogenous surfactant reduces mortality, pulmonary air leaks, duration of MV and VILI, especially in ELGANs [71].

The most recent European Consensus Guidelines on the Management of RDS does not recommend prophylactic surfactant for the smallest babies at birth [5].

As previously described, current evidence recommends early stabilization with CPAP of at least 6 cm H_2_O, via mask or nasal cannulas, in spontaneously breathing preterm infants, and early rescue surfactant administration in preterm infants with RDS. It has been demonstrated that in preterm infants (especially ELGANs) requiring surfactant therapy, earlier treatment, before 2 h of life, has benefits over later treatment.

If intubation is required as part of stabilization (i.e., in the delivery room), current evidence suggests to administer surfactant immediately, as the main purpose of avoiding surfactant prophylaxis is to avoid intubation [5,123].

#### 6.1.1. Indications for Surfactant Administration in Preterm Infants with RDS

Several studies have reported that a FiO_2_ ≥ 0.30 in the first hours after birth in infants undergoing CPAP is a reasonably good test for predicting subsequent CPAP failure [5].

Nowadays, guidelines recommended the threshold of FiO_2_ > 0.30 on CPAP pressure of at least 6 cm H_2_O for all preterm infants with a clinical diagnosis of RDS, especially in the early phase of worsening disease [124]

The need to find new criteria for surfactant administration in preterm infants with RDS, with respect to the current indications of FiO_2_ > 0.30 on CPAP pressure of at least 6 cm H_2_O, is emerging in the literature to individualize surfactant treatment as much as possible and optimize it concomitantly with other respiratory interventions.

The use of lung ultrasound (LUS) is increasingly popular, as it can define the need for surfactant before an increase in FiO_2_ needs, with the advantage of being a noninvasive procedure performed at the bedside without adverse events [124].

Bedside LUS performed by experts may be helpful in the early identification of preterm infants with RDS who require surfactant therapy. Further studies are required to confirm this.

Recently, several studies have addressed the use of new rapid bedside tests (i.e., click test, lamellar body count, stable microbubble test on a tracheal aspirate or a gastric aspirate specimen) to potentially supplement the use of clinical criteria in selecting preterm infants with RDS needing surfactant therapy [125,126]. Further research is required to elucidate the role of these tests.

Surfactant may become rapidly metabolized and functionally inactivated. The ability to administer repeated or successive doses of surfactant is thought to be helpful in overcoming such inactivation.

Based on current evidence, it appears appropriate to use persistent or worsening signs of RDS as criteria for retreatment with surfactant. The use of a higher threshold for retreatment appears to be as effective as a low threshold. A low threshold for repeat dosing should be used in preterm infants with RDS who have perinatal depression or infection [71].

Three types of exogenous surfactant are available:Surfactant derived from animal sources;Synthetic surfactant without protein components;Synthetic surfactant with protein components.

Animal-derived surfactants have greater benefits than synthetic surfactants in the treatment of RDS. Based on the best knowledge available, there is a survival advantage using 200 mg/kg of poractant alfa for the first dose and 100 mg/kg of poractant alfa for subsequent doses in the treatment of RDS [127].

#### 6.1.2. Strategy of Surfactant Administration in Preterm Infants with RDS

To date, the optimal surfactant administration method and strategy in preterm infants with RDS remain unresolved especially with the clinical focus on avoiding intubation and MV.

In intubated newborns, surfactant should be administered through a catheter inserted into the endotracheal tube (preferably closed system).

The known clinical trials evaluating surfactant administration in spontaneously breathing preterm infants with RDS have used tracheal intubation, bolus administration with distribution of surfactant using intermittent PPV, either manually or with a ventilator, followed by a period of weaning from MV as lung compliance improves.

The most recent strategy of intubation, surfactant administration, followed immediately (within 1 h) by extubation to nasal CPAP called the INSURE (INtubate, SURfactant, Extubate) technique results in brief periods of MV [128].

Although beneficial in clinical practice, the INSURE method is not always universally effective and has a failure rate in premature infants, especially ELGANs, ranging from 19% to 69% [129]. Risk factors for failure of INSURE are: low BW, low GA, the severity of initial respiratory disease, and a low hemoglobin concentration prior to surfactant administration [130].

A recent RCT showed that the application of a recruitment manoeuver in HFOV just before surfactant administration, followed by rapid extubation (Intubate-RECruit-SURfactant-Extubate (INRECSURE)), decreased the need for MV during the first 72 h of life compared with the INSURE technique in ELGANs, without increasing the risk of adverse neonatal outcomes. This reduction in the need for MV is likely due to an improved surfactant response after a recruitment procedure that achieved and maintained an “optimal” EELV more effectively than with the INSURE procedure without prior recruitment [131]. INRECSURE represents a promising novel alternative to the classical INSURE method in ELGANs.

In the last decade, new less invasive techniques for surfactant administration have been investigated in order to avoid or minimize trauma related to endotracheal intubation and MV.

These techniques have been variably called less invasive surfactant administration (LISA) or minimally invasive surfactant therapy (MIST). A thin catheter or plastic tube is inserted in the trachea under direct or video-laryngoscopy while the patient is breathing spontaneously assisted by noninvasive respiratory support, most of the times with either nasal CPAP or NIPPV, and surfactant is squirted into the airway [132].

The popularity of the LISA technique has increased because it potentially combines the benefits of early surfactant treatment with nCPAP and consequent avoidance of MV.

The most recent European Consensus Guidelines on the management of RDS recommend LISA as the preferred mode of surfactant administration in spontaneously breathing preterm infants with RDS [5].

These guidelines are based on interpretations of RCTs and meta-analyses suggesting that LISA reduces the need of endotracheal intubation and MV and is associated with lower risks of death and BPD as compared with administration of surfactant after intubation and MV.

The recommendation was considered weak because some of the background studies were open to bias, and only two of the studies that these guidelines were based on included infants with GA between 23 and 28 weeks [133,134].

A recent large multicenter RCT of infants with GA between 25 and 28 weeks did not find a significant reduction in the composite outcome of death and BPD when using minimally invasive surfactant therapy via a thin catheter compared with sham treatment [135].

In the last network meta-analyses on the comparative efficacy of methods for surfactant administration, data for preterm infants with GA < 28 weeks were not as robust as for the higher GA groups due to a smaller number of neonates [136,137].

The later updated Cochrane review confirmed the findings reported in the last European guidelines and found that LISA reduced the risk of severe intracranial hemorrhage [138].

However, the Cochrane authors identified several factors that could influence the safety and efficacy of the interventions examined: (1) the meta-analysis did not allow reliable subgroup analyses and the number of infants with GA < 28 weeks was limited; (2) ambiguity in the use of sedation and analgesia pre-medication in the tracheal catheterization group.

A recent propensity score-matched national study in a restricted analyses to GA between 25 and 27 weeks suggested that the use of LISA may reduce the need for and duration of MV in these infants, but it has little or no effect on reducing mortality or pulmonary, neurological, or gastrointestinal morbidity [139].

Other authors have also pointed out that there is insufficient evidence that LISA is preferable to surfactant administration after intubation [140].

Therefore, the safety and efficacy of LISA in this population remain to be confirmed, also considering that extreme prematurity is an independent risk factor for LISA failure [141].

A new international multicenter RCT (“INREC-LISA”) will start in 2023 and will compare the INRECSURE with LISA to evaluate the comparative efficacy of these techniques in increasing BPD-free survival in ELGANs (NCT05711966).

Moreover, marked variations in the use of LISA and outcomes between NICUs may indicate that individual NICU experience may be more important than a LISA versus a non-LISA strategy. GA < 26 weeks could be considered a contraindication to LISA in inexperienced hands [142].

Other types of less invasive approaches to delivering surfactant in preterm infants with RDS are described, including aerosolization, pharyngeal instillation and laryngeal mask administration. However, no solid data are available with regard to ELGAN.

Atomized surfactant delivered by nebulization would be truly noninvasive, although only one recent clinical trial has shown that nebulizing surfactant when on CPAP reduces the need for MV compared to CPAP alone, and this finding was limited to a subgroup of more mature preterm infants (GA 32–33 weeks) [143].

There are many challenges that will need to be overcome, such as dosing, duration of aerosolization, and potential loss of surfactant to the upper airway.

Surfactant administration through a laryngeal mask airway (LMA) is noninvasive, avoids endotracheal intubation, and potentially avoids the complications associated with intubation [144].

However, the use of the currently available LMA is recommended only for newborns above 1500 g. Moreover, the pharyngeal deposition of surfactant at birth is currently being tested and several adverse events are reported with the use of LMA. Further research is required to establish the efficacy and risk-versus-benefit ratio of these methods [145].

More studies are required before firm conclusions can be drawn about the optimal method of administration of surfactant and whether the optimal method is different for different types of preterm infants.

Further well-designed studies of adequate size and power, as well as ongoing studies, will help confirm and refine these findings, clarify whether surfactant therapy via thin tracheal catheter provides benefits over other techniques of surfactant administration, address uncertainties within important subgroups, and clarify the role of sedation.

### 6.2. Surfactant Therapy in Preterm Infants with Other Respiratory Disorders

Exogenous surfactant therapy has been attempted in a variety of neonatal preterm respiratory disorders other than RDS, with variable quality of evidence and variable efficacy [146].

Data from animal studies and infants with evolving or with established BPD have demonstrated quantitative and qualitative abnormalities of surfactant.

Observational studies have shown transient improvements in oxygenation and ventilatory support among infants with BPD given exogenous surfactant. However, no major impact on prevention of BPD has been reported to date.

Therefore, administration of surfactant alone for infants evolving to BPD remains under study and cannot be widely recommended [122].

Using surfactant as a vehicle for other interventions, i.e., inhaled budesonide, aimed at preventing BPD may become a viable alternative to systemic postnatal corticosteroids. Much of this evidence has been reviewed recently [147].

However, which infants should receive exogenous surfactant as a medium to deliver budesonide directly to the airspaces and the timing and dosing of this intervention remain in question before this can be recommended [148]. The results of a large RCT (NCT04545866) will be published shortly.

There are some reports concerning the use of exogenous surfactant therapy in the management of pulmonary hemorrhage and pneumonia in preterm infants, but its efficacy in these conditions is uncertain without evidence recommendations [71].

Data with variable quality of evidence are present about exogenous surfactant therapy in other neonatal respiratory disorders such as meconium aspiration syndrome and congenital diaphragmatic hernia [71].

## 7. Results

Knowing the limitations of this narrative and non-systematic review, we tried to summarize the principal current evidence on the respiratory management of the preterm infants in Table 3.

## 8. Discussion

Respiratory failure represents one of the most critical clinical conditions affecting preterm infants, inversely to their GA. Adequate respiratory management of preterm infants, especially ELGANs (<28 weeks of GA), is essential to ensuring adequate gas exchange, reducing VILI, promoting growth, and reducing overall neonatal morbidity and mortality.

Neonatologists today must implement the best evidence-based practice for respiratory support at the bedside, starting from the delivery room and extending throughout the patient’s entire NICU stay. This assistance must be tailored to the specific characteristics and timing of the treated pulmonary disorder.

By following an imaginary timeline proceeding from the first respiratory assistance provided at birth, and continuing throughout a preterm infant’s NICU stay, with this review, we have summarized the most important evidence-based practice recommendations currently available for the respiratory management of the preterm infant.

Current evidence suggests the primary use of noninvasive respiratory support in spontaneously breathing preterm infants to avoid MV and thus reduce morbidities and mortality. Nevertheless, a substantial portion of preterm infants require lifesaving intubation and MV in the delivery room and/or at some point during their NICU stay.

For these reasons, neonatologists must be able to manage both noninvasive and invasive respiratory support systems for treating a spectrum of lung diseases ranging from acute to chronic conditions.

Tendency to atelectasis, inability to establish an FRC, hypoxemia, hypercapnia, and increased work of breathing are the clinical characteristics that describe respiratory failure of preterm infants when initiating breathing. Recruitment maneuvers and exogenous surfactant administration play a key role in counteracting these mechanisms. The positive effects of recruitment can be achieved: (a) in the delivery room, by applying adequate CPAP levels to overcome the very high resistance component of the driving pressure, typical of a fluid-filled and surfactant-deficient lung; (b) in the NICU, by applying adequate MAP levels combined with surfactant therapy, to overcome the very high elastic component of the driving pressure, which is typical of a surfactant-deficient low-compliant lung.

## 9. Conclusions

Summarizing the current evidences about respiratory management of the preterm infants, it is possible to conclude:–In delivery room: nCPAP initiated in the delivery room compared with intubation reduces death or BPD in very preterm infants. Current European guidelines advise using CPAP of at least 6 cm H_2_O. Endotracheal intubation should be considered only for infants who do not develop adequate respiratory effort and/or who remain bradycardic and/or hypoxic despite adequate mask or nasal prongs PPV.–In the NICU, the most effective noninvasive respiratory support as primary mode is NIPPV, especially sNIPPV. As an alternative primary support, it seems appropriate to choose nCPAP in infants born <28 weeks of GA. As post-extubation respiratory support, it seems reasonable to choose NIPPV in preterm neonates at high risk of extubation failure, especially in ELGANs. In case of low risk of extubation failure, it is possible to choose nCPAP.–MV is associated with increased mortality and pulmonary and systemic morbidities. Preterm infants needing MV should be ventilated using volume-target ventilation (VTV), resulting in a reduction in the risk of BPD, hypocapnia, and other morbidities. The selection of VT or VThf may be individualized for the specific patient, considering her/his lung mechanics, lung dimension and the phase of the lung disease being treated (acute/chronic/weaning). Weaning preterm infants from MV as soon as possible represents an imperative for all neonatologists.–Nowadays, guidelines recommended administering surfactant if FiO2 is >0.30 on CPAP pressure of at least 6 cm H_2_O for all preterm infants with a clinical diagnosis of RDS. The need to find new criteria for surfactant administration is emerging in the literature. Although the most recent European consensus guidelines on the management of RDS recommend LISA as the preferred mode of surfactant administration, the level of evidence is weak, and more studies are required before firm conclusions can be drawn about the optimal method of administration of surfactant.

## Figures and Tables

**Figure 1 children-10-00535-f001:**
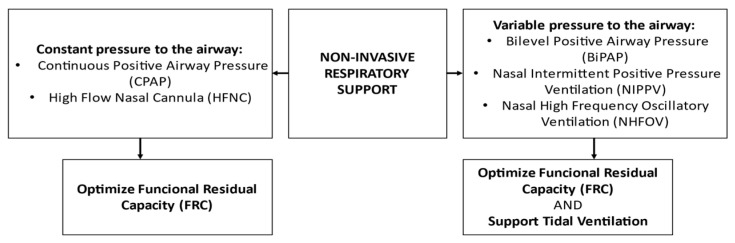
Different non-invasive respiratory support techniques.

**Figure 2 children-10-00535-f002:**
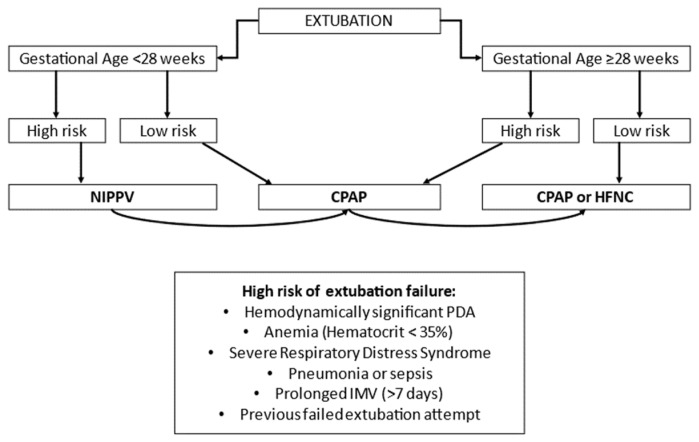
Proposed algorithm for choosing post-extubation noninvasive support modality. NIPPV: Nasal Intermittent Positive Pressure Ventilation, CPAP: Nasal continuous positive airway pressure, HFNC: High-Flow Nasal Cannula, PDA: Patent ductus arteriosus, IMV: Invasive mechanical ventilation.

**Table 1 children-10-00535-t001:** Suggested indications for intubation and mechanical ventilation in preterm infants: presence of at least one of the following criteria.

Criteria	Description
Excessive work of breathing	Dyspnea with Silverman score > 6 and/or severe tachypnea (>100 breaths/min), despite optimized noninvasive respiratory support
Absent or inadequate respiratory effort	Apnea > 4 events/hour or >2 events/hour requiring positive pressure ventilation with mask, despite optimized noninvasive respiratory support and adequate caffeine therapy
Severe respiratory acidosis	Arterial/Capillary blood: pH < 7.20 and pCO_2_ > 60 mmHg at 0–72 h of life, pCO_2_ > 65 mmHg beyond 72 h of life, despite optimized noninvasive respiratory support
High oxygen requirement	FiO_2_ > 0.50 for ELGANs or FiO_2_ > 0.60 for newborns between 28 and 32 weeks of GA, to maintain adequate range value of paO_2_ > 50–60 mmHg (6.7–8 kPa) and adequate range of SpO_2_ (90–95%) despite optimized noninvasive respiratory support and surfactant treatment for RDS
Moderate or severe respiratory distress and contraindications for noninvasive ventilatory support	Intestinal perforation, Intestinal obstruction, esophageal atresia Recent gastrointestinal surgery
Postoperative period	Recent abdominal incision Recent tracheostomy Residual effects of anesthetic agents Need for muscle relaxant drugs

pCO_2_: partial pressure of CO_2_; FiO_2_: fraction of inspired oxygen; ELGANs: extremely low gestational age newborns; paO_2_: arterial oxygen tension; RDS: respiratory distress syndrome.

**Table 2 children-10-00535-t002:** Suggested goals of mechanical ventilation in preterm infants during NICU stay.

Value	Preterm Infants	Infants with BPD	Infants with BPD and PPHN
pH (arterial)	7.25–7.35	≥7.25	≥7.25
PaO_2_ (mmHg)	45–65	45–65	55–75
PaCO_2_ (mmHg): ◆ Goal ◆ Tolerated 0–72 h of life ◆ Tolerated > 72 h of life	45–55 <60 <65	55–65 <70	45–60 <70
SpO_2_ (%)	90–95	92–95	97–98

PaO_2_: partial pressure of O_2;_ PaCO_2_: partial pressure of CO_2_; SpO_2_: preductal oxygen saturation.

**Table 3 children-10-00535-t003:** Summary of current evidence for respiratory management of preterm infants.

Phase	Respiratory Management
Delivery room stabilization	Early initiation of CPAP Noninvasive respiratory support in spontaneously breathing infants to avoid intubation
	Use of T-piece resuscitators Use of oxygen blender Preductal SpO_2_ > 80% by 5 min of life Target SpO_2_: 90–95% Avoid prolonged period of hypoxia (SpO_2_ < 80%) and fluctuation in SpO_2_
Noninvasive respiratory support in Neonatal Intensive Care Unit	Encourage noninvasive respiratory support (CPAP, NIPPV, SNIPPV) avoiding endotracheal intubation and mechanical ventilation (see Figure 1 and Figure 2) Avoid prolonged period of hypoxia (SpO_2_ < 80%) and fluctuation in SpO_2_
Mechanical ventilation in Neonatal Intensive Care Unit	Refer to specific indications for intubation and mechanical ventilation (see Table 1)
	Refer to specific goals of MV Avoid prolonged period of hypoxia (SpO_2_ < 80%) and fluctuation in SpO_2_ Choose lung protective ventilation both during CMV and HFOV Consider volume target ventilation strategy both during CMV and HFOV Refer to specific weaning and extubation criteria (i.e., clinical stability, MAP < 8 cm H_2_O and FiO_2_ < 30%) Trial of extubation to CPAP/NIPPV/SNIPPV prior to 7 days of life or as early as possible
Surfactant administration	Administer surfactant as early as possible in preterm infants with RDS that require FiO_2_ > 30% in CPAP ≥ 6 cm H_2_O

## Data Availability

Not applicable.

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
