# Peer review of "Respiratory Management of the Preterm Infant: Supporting Evidence-Based Practice at the Bedside"

_children, 2023, doi:10.3390/children10030535_

Round 1
Reviewer 1 Report
Dear Authors,
Respiratory management of the preterm infant is an important subject to discuss and there are still no valid universal management guidelines. All the more, your review is welcome.
I have some minor comments.
-Please revise English language and typo; the sentences are too long.
-Line 112- please define RCT at the first appearance in the text (I suppose to be a randomized controlled trial)
-Line 138-142 and 148-156 - References are needed here
-Lines 368-384 are redundant, as they are the same with Table 1
There is any limitation of the study?
Author Response
Please revise English language and typo; the sentences are too long.
- Thank you for your comments. We revised English language, editing it by a native english speaker and modified the length of sentences in the text, as you suggested.
Line 112- please define RCT at the first appearance in the text (I suppose to be a randomized controlled trial)
- Line 112- we defined RCT in the first appearance as randomized controlled trial, as you suggested.
Line 138-142 and 148-156 - References are needed here
- we added references as you suggested :
- 138-142 [20-23]
- 148-156 [24]
-ines 368-384 are redundant, as they are the same with Table 1
- Lines 368-384 We deleted sentence in these lines to avoid redundancy with Table 1
There is any limitation of the study?
The main limitation of the study is the narrative and non-systematic nature of the review and we specified it in the text..
Reviewer 2 Report
The manuscript summarized the current evidence on the respiratory management of preterm infants. This review is interesting but still not well-written. Some major and minor comments are given below. I am happy to re-review it in the next round.
Major comments:
1. Methods: There is no methods section in this review. The authors do not present any information on the methods they used to summarize the evidence. Please add this important section otherwise it should not be considered for publication. Specifically:
• Please describe the search strategy with sufficient information.
• What type of review?
• Did you include any Medical Subject Headings (MeSH) terms?
• Please provide the PRISMA Flow Diagram to map out the number of records identified, included, and excluded, and the reasons for exclusions.
http://prisma-statement.org/prismastatement/flowdiagram.aspx
2. Abstract
• The abstract is not well-written. It only presents the background and objective of the study. How about the methods, results, and conclusions?
3. Introduction
• The introduction needs to be improved. Some important information should be added such as the global burden of respiratory failure among preterm infants, the current global and/or national guidelines, and the current gap that needs to conduct this study (rationality).
4. Results
• Please provide a theoretical framework so that readers can understand the flow of this study.
• Please re-construct the manuscript so that all the results should be in one section.
• As the authors stated that they aimed to provide “a practical guide to the neonatologist”, they should present a clear and usable guideline either in results or supplement.
5. Discussion
• There is no discussion section in this paper. Please provide one with sufficient information on summary findings, discuss main findings with a comparison with previous works, implications for policy and practice, strengths and limitations, etc.
Author Response
Comments and Suggestions for Authors:
The manuscript summarized the current evidence on the respiratory management of preterm infants. This review is interesting but still not well-written. Some major and minor comments are given below. I am happy to re-review it in the next round.
Major comments:
- Methods: There is no methods section in this review. The authors do not present any information on the methods they used to summarize the evidence. Please add this important section otherwise it should not be considered for publication. Specifically:
- Please describe the search strategy with sufficient information.
- What type of review?
- Did you include any Medical Subject Headings (MeSH) terms?
- Please provide the PRISMA Flow Diagram to map out the number of records identified, included, and excluded, and the reasons for exclusions.
http://prisma-statement.org/prismastatement/flowdiagram.aspx
- Reply :
Thank you for your important comment.
We didn’t include a methods section in the first version of our narrative review because we followed the Journal’s indications. Considering that the type of article we submitted to the Journal is considered a “Review”, we followed the definition found on the website (https://www.mdpi.com/about/article_types): “…the structure can include an Abstract, Keywords, Introduction, Relevant Sections, Discussion, Conclusions, and Future Directions…”
Furthermore, a detailed investigation of previous research on a given topic with clearly defined search parameters, following the PRISMA checklist and flow diagram as you suggested, is recommended for a “Systematic Review” type article in the website (https://www.mdpi.com/about/article_types).
However, accepting your suggestion, we have introduced a methods section in this new version:
- Methods
In order to identify papers considered in this narrative review, we performed a literature search on PubMED. The research has been restricted to papers in English language. We limited the search by applying the filter of age “ preterm infant” and used the following search terms and logic: “preterm infant AND respiratory management in the delivery room”, “preterm infant AND non-invasive ventilation,” “preterm infant AND respiratory support”, “preterm infants AND nasal continuous positive airway pressure”, “preterm infants AND high flow nasal cannula”, “preterm infant AND heated humidified high flow nasal cannula”, “preterm infant AND Nasal Intermittent Positive Pressure Ventilation”, “ preterm infants AND mechanical ventilation”, “ preterm infants AND high frequency oscillatory ventilation”, “ preterm infants AND Lung-Protective Ventilation”, “ preterm infants AND optimal lung volume strategy”, “ preterm infants AND Volume-Targeted Ventilation”, “ preterm infants AND Weaning from mechanical ventilation”, “preterm infants AND and extubation from mechanical ventilation”, “preterm infants AND exogenous surfactant therapy”.
No limit about year of publication has been set, and the final search is updated to December 2022. To identify any articles that may have been missed during the literature search, also reference lists of candidate articles have been carefully checked.
Moreover, we included information from the last edition (7th) of Goldsmith’s Assisted Ventilation of the neonate.
.
- Abstract
- The abstract is not well-written. It only presents the background and objective of the study. How about the methods, results, and conclusions?
Thank you for your comment.
We have rewritten the Abstract and the language was edited by a native English speaker. The methods were also added to the Abstract.
As reported in the previous point, results probably do not fit with the type of article intended for submission to the journal, considering the fact that it is a “Review” as defined in the website (https://www.mdpi.com/about/article_types); for this reason we did not report the results in the Abstract.
Regarding the conclusions in the abstract we have considered the manuscript an evidence-based overview regarding respiratory management of preterm infants, especially in the acute phase of neonatal respiratory distress syndrome, starting from the delivery room and continuing in the neonatal intensive care unit, including a section regarding surfactant therapy.
- Introduction
- The introduction needs to be improved. Some important information should be added such as the global burden of respiratory failure among preterm infants, the current global and/or national guidelines, and the current gap that needs to conduct this study (rationality).
Thank you for your comment.
As you suggested we improved the introduction and added epidemiological data and information about available guidelines. As specified in the text, the aim of this narrative review would be to summarize the current evidence on the respiratory management of preterm infants, providing a practical guide for the neonatologist, especially in the early phase of neonatal respiratory distress syndrome (RDS).
- Results
- Please provide a theoretical framework so that readers can understand the flow of this study.
- Please re-construct the manuscript so that all the results should be in one section.
- As the authors stated that they aimed to provide “a practical guide to the neonatologist”, they should present a clear and usable guideline either in results or supplement.
Thank you for your suggestion.
The aim of this narrative review is to summarize the current evidence on the respiratory management of preterm infants providing a practical guide.
At the end of the Introduction we have added a list of the relevant sections discussed to better understand the flow of our narrative review.
“The aim of this narrative review would be to summarize the current evidence on the respiratory management of the preterm infants providing a practical guide to the neonatologist, especially in the early phase of neonatal respiratory distress syndrome (RDS) including these relevant sections :
- Respiratory management of the preterm infant in the Delivery Room
- Noninvasive respiratory support of the preterm infants in the neonatal intensive care unit
- Mechanical ventilation of the preterm infants in the neonatal intensive care unit
- Exogenous Surfactant therapy in preterm infants”
There was no Results section in the first version of our narrative review because we followed the Journal’s indications, considering the type of article submitted to the Journal is a “Review” as defined in the website (https://www.mdpi.com/about/article_types): “…the structure can include an Abstract, Keywords, Introduction, Relevant Sections, Discussion, Conclusions, and Future Directions…”
As you suggested, we have introduced a Results section in this new version.
- Results
Knowing the limitations of this narrative and non-systematic review, we tried to summarize the principal current evidence on the respiratory management of preterm infants in Table 3.
Table 3. Summary of current evidence for respiratory management of preterm infants
Phase |
Respiratory Management |
Delivery Room stabilization |
Early initiation of CPAP Non invasive respiratory support in spontaneously breathing infants to avoid intubation |
Use of T-piece resuscitators Use of Oxygen blender Preductal SpO2 > 80% by 5 minutes of life Target SpO2 : 90-95% Avoid prolonged period of hypoxia (SpO2 < 80%) and fluctuation in SpO2 |
|
|
|
Non Invasive respiratory support in Neonatal Intensive Care Unit |
Encourage non invasive respiratory support (CPAP, NIPPV, SNIPPV) avoiding endotracheal intubation and mechanical ventilation (See Figure 1 and Figure 2) Avoid prolonged period of hypoxia (SpO2 < 80%) and fluctuation in SpO2 |
Mechanical Ventilation in Neonatal Intensive Care Unit |
Refer to specific indications for intubation and mechanical ventilation (see Table 1 ) |
Refer to specifica goals of MV Avoid prolonged period of hypoxia (SpO2 < 80%) and fluctuation in SpO2 Choose Lung protective Ventilation both during CMV and HFOV Consider volume target ventilation strategy both during CMV and HFOV Refer to specific weaning and extubation criteria (i.e clinical stability, MAP< 8 cmH2O and FiO2 < 30%) Trial of extubation to CPAP/NIPPV/SNIPPV prior to 7 days of life or as early as possible |
|
Surfactant administration |
Administer surfactant as early as possible in preterm infants with RDS that require FiO2>30% in CPAP ≥6 cmH2O |
|
CPAP: continuous positive airway pressure; NIPPV: Nasal Intermittent Positive Pressure Ventilation; SNIPPV: Synchronized Nasal Intermittent Positive Pressure Ventilation; ”, CMV: conventional mechanical ventilation”, HFOV: high frequency oscillatory ventilation; MAP: Mean Airway Pressure, RDS: Respiratory Distress Syndrome
- Discussion
- There is no discussion section in this paper. Please provide one with sufficient information on summary findings, discuss main findings with a comparison with previous works, implications for policy and practice, strengths and limitations, etc.
Thanks for your comment. We added discussion section and modified the conclusion section.
- Discussion
Respiratory failure represents one of the most critical clinical conditions affecting preterm infants, inversely to their GA. Adequate respiratory management of preterm infants, especially ELGANs (< 28 weeks of GA), is essential to ensuring adequate gas exchange, reducing VILI, promoting growth, and reducing overall neonatal morbidity and mortality.
Neonatologists today must implement the best evidence-based practice for respiratory support at the bedside, starting from the delivery room and extending throughout the patient’s entire NICU stay. This assistance must be tailored to the specific characteristics and timing of the treated pulmonary disorder.
By following an imaginary timeline proceeding from the first respiratory assistance provided at birth, and continuing throughout a preterm infant’s NICU stay, with this review we have summarized the most important evidence-based practice recommendations currently available for the respiratory management of the preterm infant.
Current evidence suggests primary use of noninvasive respiratory support in spontaneously breathing preterm infants, to avoid MV and thus reduce morbidities and mortality. Nevertheless, a substantial portion of preterm infants require lifesaving intubation and MV in the delivery room and/or at some point during their NICU stay.
For these reasons, neonatologists must be able to manage both noninvasive and invasive respiratory support systems for treating a spectrum of lung diseases ranging from acute to chronic conditions.
Tendency to atelectasis, inability to establish a FRC, hypoxemia, hypercapnia, and increased work of breathing are the clinical characteristics that describe respiratory failure of preterm infants when initiating breathing. Recruitment maneuvers and exogenous surfactant administration play a key role in counteracting these mechanisms. The positive effects of recruitment can be achieved: a) in the Delivery room, by applying adequate CPAP levels to overcome the very high resistance component of the driving pressure, typical of a fluid-filled and surfactant-deficient lung; b) in the NICU, by applying adequate MAP levels combined with surfactant therapy, to overcome the very high elastic component of the driving pressure, typical of a surfactant-deficient low-compliant lung.
- Conclusion
Summarizing the current evidences about respiratory management of the preterm infants, it’s possible to conclude to:
- In delivery room: nCPAP initiated in the delivery room compared with intubation reduces death or BPD in very preterm infants. Current European guidelines advise using CPAP of at least 6 cm H2O. Endotracheal intubation should be considered only for infants who do not develop adequate respiratory effort and/or who remain bradycardic and/or hypoxic despite adequate mask or nasal prongs PPV.
- In NICU, the most effective non-invasive respiratory support as primary mode is NIPPV, especially sNIPPV. As an alternative primary support, it seems appropriate to choose nCPAP in infants born < 28 weeks of GA. As post extubation respiratory support, it seems reasonable to choose NIPPV in preterm neonates at high risk of extubation failure, especially in ELGANs. In case of low risk of extubation failure, it’s possible to choose nCPAP.
- MV is associated with increased mortality and pulmonary and systemic morbidities. Preterm infants needing MV should be ventilated using volume-target ventilation (VTV), resulting in a reduction in the risk of BPD, hypocapnia, and other morbidities. The selection of VT or VThf may be individualized for the specific patient, considering her/him lung mechanics, lung dimension and the phase of the lung disease being treated (acute/chronic/weaning). Weaning preterm infants from MV as soon as possible represents an imperative for all neonatologists.
- Nowadays guidelines recommended administering surfactant if FiO2 is > 0.30 on CPAP pressure of at least 6 cm H2O for all preterm infants with a clinical diagnosis of RDS. The need to find new criteria for surfactant administration is emerging in the literature. Although the most recent European Consensus Guidelines on the management of RDS recommends LISA as the preferred mode of surfactant administration, the level of evidence is weak and more studies are required before firm conclusions can be drawn about the optimal method of administration of surfactant.
Reviewer 3 Report
The paper is very clearly organized, with a wide range of recent research also appropriately presented. It will be useful not only to physicians, but also to nurses working in NICUs and medical students.
Author Response
The paper is very clearly organized, with a wide range of recent research also appropriately presented. It will be useful not only to physicians, but also to nurses working in NICUs and medical students.
- Thanks for your valuable comments
Reviewer 4 Report
Dear authors, you conducted an excellent review regarding the respiratory support of preterm neonates.
I find it very usefull for Neonatologists.
In line 20 "i.e." is not a very appropriate term for an abstract. Please rephrase.
The tables and provided algorithms are very helpfull for the clinical practice.
Nevertheless, the english are not very good. I suggest a language edit by a naitve english speaker.
Author Response
Dear authors, you conducted an excellent review regarding the respiratory support of preterm neonates.
I find it very usefull for Neonatologists.
In line 20 "i.e." is not a very appropriate term for an abstract. Please rephrase.
- Thank you for your valuable comments. As you suggested we modified line 20 deleting “i.e.” and modifing the sentence : “to ensure an adequate functional residual capacity”
The tables and provided algorithms are very helpfull for the clinical practice.
- Thank you for your comment
Nevertheless, the english are not very good. I suggest a language edit by a naitve english speaker.
- As you suggested we revised english editing language by a native english speaker